# Development of a Nucleocapsid Protein-Based Blocking ELISA for the Detection of Porcine Deltacoronavirus Antibodies

**DOI:** 10.3390/v14081815

**Published:** 2022-08-18

**Authors:** Wenlong Wang, Yongning Zhang, Hanchun Yang

**Affiliations:** Key Laboratory of Animal Epidemiology of Ministry of Agriculture and Rural Affairs, College of Veterinary Medicine, China Agricultural University, Beijing 100193, China

**Keywords:** porcine deltacoronavirus (PDCoV), nucleocapsid protein, blocking ELISA, detection, antibodies

## Abstract

Porcine deltacoronavirus (PDCoV) is an emerging enteropathogen which mainly causes diarrhea, dehydration and death in nursing piglets, threatening the global swine industry. Moreover, it can infect multiple animal species and humans. Hence, reliable diagnostic assays are needed to better control this zoonotic pathogen. Here, a blocking ELISA was developed using a recombinant nucleocapsid (N) protein as the coating antigen paired with an N-specific monoclonal antibody (mAb) as the detection antibody. The percent inhibition (PI) of the ELISA was determined using 384 swine serum samples, with an indirect immunofluorescence assay (IFA) as the reference method. Through receiver operating characteristic analysis in conjunction with Youden’s index, the optimal PI cut-off value was determined to be 51.65%, which corresponded to a diagnostic sensitivity of 98.79% and a diagnostic specificity of 100%. Of the 330 serum samples tested positive via IFA, 326 and 4 were tested positive and negative via the ELISA, respectively, while the 54 serum samples tested negative via IFA were all negative via the ELISA. The overall coincidence rate between the two assays was 98.96% (380/384). The ELISA exhibited good repeatability and did not cross-react with antisera against other swine pathogens. Overall, this is the first report on developing a blocking ELISA for PDCoV serodiagnosis.

## 1. Introduction

Porcine deltacoronavirus (PDCoV) is a new type of porcine enteropathogen which mainly infects piglets and causes severe diarrhea, vomiting, dehydration, emaciation and eventual death [1]. Pigs of other ages, including sows, are also susceptible to PDCoV infection, making it one of the most crucial pathogens affecting the global pig industry [1,2,3].

In 2012, Woo et al. identified the complete genome sequence of PDCoV for the first time from pig rectal swabs which were collected during 2009–2010 [4]. In 2014, Wang et al. obtained the complete genome sequence of PDCoV from fecal and intestinal contents of pigs suffering from diarrhea in Ohio, USA [5]. Subsequently, Hu et al. successfully isolated PDCoV strain OH-FD22 from the intestines of diarrhea-afflicted pigs using LLC-PK1 cells and confirmed that PDCoV has been widely prevalent in pig populations in multiple states of the USA [6]. So far, PDCoV infection has been reported in Canada [7], China [8], South Korea [9], Thailand [10], Laos [11], Vietnam [12], Japan [13], Mexico [14] and so on. In addition to pigs, PDCoV can also infect various animal species such as wild birds, chickens, turkeys, cattle and calves, indicating that PDCoV has a potential risk of cross-species transmission [15]. In 2021, researchers successfully isolated PDCoV from the plasma of three Haitian children suffering from fever and abdominal pain [16], thereby confirming the public health significance of PDCoV.

PDCoV is a single-stranded, positive-sense RNA virus with an envelope. It is classified in the genus *Deltacoronavirus* within the *Coronaviridae* family [4]. The full length of the PDCoV genome is about 25.4 kb, consisting of a 5’ untranslated region (5’UTR), open reading frame 1a/1b (ORF1a/1b), spike protein (S), envelope protein (E), membrane protein (M), nonstructural protein 6, nucleocapsid protein (N), nonstructural protein 7, 3’UTR and poly (A) tail [17,18]. As a phosphorylated protein, the N protein of PDCoV binds with viral genomic RNA to form a nucleocapsid, which is required to constitute the core of the virus together with the M protein [19,20]. During PDCoV infection in vivo, the N protein is highly expressed in PDCoV-infected host cells, thereby stimulating the body to produce a high level of PDCoV-specific antibodies. More importantly, it is highly conserved among distinctive PDCoV strains [21]. Hence, the N protein is widely used as the diagnostic target of PDCoV at both the RNA and protein levels.

At present, various molecular biological detection methods, such as N or M gene-based reverse transcription PCR (RT-PCR), real-time RT-PCR, insulated isothermal RT-PCR and recombinase polymerase amplification have been developed to diagnose PDCoV infection [22]. These methods can detect PDCoV nucleic acid in diverse swine samples, such as feces, rectal swabs, intestines, sera and tissue samples [22]. As for serological detection methods for PDCoV, the indirect immunofluorescence assay (IFA) [18,21], the virus neutralization test or fluorescent focus neutralization test [23], the fluorescent microsphere immunoassay [21] and several kinds of ELISA [21,24,25,26] have been established and applied to detect PDCoV infection. The available ELISA tests include recombinant N protein-based indirect ELISA [21,27], recombinant S1 protein-based indirect ELISA [25], recombinant M protein-based indirect ELISA [24], and recombinant N protein-based double antibody sandwich ELISA [26], the first three of which are mainly used for the detection of PDCoV antibodies, whereas the latter is suitable for the detection of PDCoV antigens. Although each of these various types of ELISA tests has played an important role in the diagnosis of PDCoV, they are not suitable for the detection of samples of animal species other than pigs. As mentioned above, PDCoV has the ability of interspecific transmission and can infect a variety of animals and humans. If the foregoing ELISA tests are to be used for the diagnosis of PDCoV infection in animal species other than pigs, horseradish peroxidase (HRP)-conjugated secondary antibodies of different animal species are needed to match the animal species to be tested. Obviously, this shortcoming limits the application of these ELISA tests in PDCoV epidemiological investigations.

In the present study, in order to establish an improved ELISA with a wider application scope and higher diagnostic specificity, we developed a blocking ELISA on the basis of the preparation of a recombinant PDCoV N protein and its specific monoclonal antibody (mAb). The developed assay is able to detect PDCoV-specific antibodies without being limited by animal species. Moreover, we compared the diagnostic performance of our developed blocking ELISA in clinical sample detection with that of an IFA test, which was used as the reference method and was performed in parallel.

## 2. Materials and Methods

### 2.1. Viruses, Cells, Antibodies and Serum Samples

PDCoV strain CHN-HN-1601 (GenBank accession No. MG832584.1) was previously isolated from the fecal sample of a diarrheic piglet in a commercial pig farm in Henan Province, China [8]. Its whole genome sequence shared 97.5–99.5% nucleotide identities with other representative PDCoV strains from multiple countries [8]. Porcine enteric alphacoronavirus (PEAV; GenBank accession No. MW727454.1) and porcine epidemic diarrhea virus (PEDV; GenBank accession No. KX066126.1) were isolated and preserved in our laboratory. Transmissible gastroenteritis virus (TGEV) strain Jms-infected ST cells and porcine rotavirus (PRoV) strain OSU-infected MA104 cells grown in 96-well cell culture plates and fixed with ethanol were generously provided by Prof. Pinghuang Liu at the College of Veterinary Medicine (China Agricultural University, Beijing, China).

Pig kidney epithelial (LLC-PK1), Vero, ST, and MA104 cells were cultured in Dulbecco’s Modified Eagle’s Medium (DMEM; Thermo Fisher Scientific, Waltham, MA, USA) supplemented with antibiotics (100 U/mL of penicillin and 100 µg/mL of streptomycin; Thermo Fisher Scientific) and 10% fetal bovine serum (FBS; Gibco, Carlsbad, CA, USA). Fluorescein isothiocyanate (FITC)-conjugated Goat anti-Mouse IgG (H + L) was purchased from Jackson ImmunoResearch Laboratories Inc. (West Grove, PA, USA). FITC-conjugated Rabbit anti-Pig IgG (H + L), HRP-conjugated Goat anti-Mouse IgG (H + L), and a PageRuler^™^ Prestained Protein Ladder were purchased from Thermo Fisher Scientific. Mouse anti-His-Tag mAb was purchased from Proteintech Group, Inc. (Rosemont, IL, USA). Nickel nitrilotriacetic acid (Ni–NTA) agarose was purchased from Qiagen (Hilden, Germany). Mouse anti-PDCoV N protein mAb 1A3 was prepared in our laboratory [8]. The antigenic epitope recognized by 1A3 was mapped to be located at a highly conserved region between amino acids 276 and 285 (RLKDALNTVV) of PDCoV N protein (Unpublished data). Rabbit anti-PDCoV N polyclonal antibody was previously prepared using the recombinant PDCoV N protein in our laboratory [28]. Swine antisera against PEDV, TGEV, PRoV, and PEAV were preserved in our laboratory. One PDCoV antibody-positive reference serum S23 (IFA titer of 1:128), one PDCoV antibody-negative reference serum S0 and three PDCoV antisera representing strongly, moderately and weakly positive samples were collected during one of our previously conducted pig challenge experiments [8]. Moreover, a total of 384 swine serum samples were collected from 28-day-old piglets experimentally infected with PDCoV strain CHN-HN-1601 during 0–63 days post-infection (dpi), and from another challenge experiment during which 5-day-old piglets were experimentally infected with PDCoV for 28 dpi [8]. The viral challenge and sample collection procedures were approved by the Laboratory Animal Welfare and Animal Experimental Ethical Committee of China Agricultural University, Beijing, China (Approval No. AW81402202-2-1).

### 2.2. Plasmid Construction

The full-length N gene (excluding the termination codon TAG) of PDCoV strain CHN-HN-1601 was amplified via PCR using the cDNA infectious clone plasmid pBeloBAC11-PDCoV we previously constructed [8] as the template, and with the specific forward (5′- CAGCAAATGGGTCGCGGATCCATGGCCGCACCAGTAGTC -3′) and reverse primers (5′- GTGGTGGTGGTGGTGGTGCTCGAGCGCTGCTGATTCCTGCTT -3′). PCR amplification was performed using the Platinum^TM^ SuperFi^TM^ II Green PCR Master Mix (Thermo Fisher Scientific) according to the manufacturer′s instructions. The resulting PCR fragment was separated and purified from 1.5% (*w*/*v*) agarose gel using an OMEGA Gel Extraction Kit (Norcross, GA, USA). The recovered amplicon was then cloned into the prokaryotic expression vector pET-28a (+) between the BamH I and Xho I sites via homologous recombination using the ClonExpress MultiS One Step Cloning Kit (Vazyme, Nanjing, China). The constructed recombinant plasmid was designated pET-28a-PDCoV-N after identification via Sanger sequencing (data not shown). The plasmid contained both an N- and a C-terminal 6× His-tag coding sequence, which would produce a His-tagged full-length PDCoV N protein.

### 2.3. Expression and Purification of His-Tagged PDCoV N Protein

The recombinant plasmid pET-28a-PDCoV-N was transformed into *E. coli* Rosetta (DE3) competent cells. The resulting positive transformants were induced with isopropyl β-D-thiogalactoside (IPTG) to express the N protein. Briefly, the positive transformants were cultured in Luria–Bertani liquid medium containing 100 μg/mL of kanamycin at 37 °C with shaking at 250 rpm and induced with 1 mM IPTG when the optical density at 600 nm for the culture reached 0.6. At 6 h post-induction, bacteria were harvested and lysed via ultrasonication (300 W, 3 s bursts with 6 s pauses) for 30 min on ice. After centrifugation (12,000 rpm, 30 min, 4 °C), both the supernatant and precipitate (inclusion bodies) of the bacterial lysates were collected and analyzed via SDS-PAGE along with Coomassie blue staining. The recombinant N protein was purified from the supernatant with Ni-NTA agarose (Qiagen) under native conditions according to the manufacturer’s instructions. The eluted protein solution was concentrated using Millipore’s Amicon Ultra-15 Centrifugal Filter Unit (Bedford, MA, USA), and the final protein concentration was determined using a Pierce BCA Protein Assay Kit (Thermo Fisher Scientific).

### 2.4. SDS-PAGE and Western Blot

Both the supernatant and precipitate of the bacterial lysates as well as the purified N protein were separated on 12% SDS-PAGE gels and then analyzed via Western blot. Briefly, the target protein in the gel was transferred onto a 0.22 µm polyvinylidene fluoride (PVDF) membrane (Millipore). After blocking with 5% skimmed milk for 2 h at room temperature, the membrane was incubated with either mouse anti-His mAb (1:10,000 dilution) or mouse anti-PDCoV N mAb 1A3 (supernatant of the hybridoma cells; 1:40 dilution) for 12 h at 4 °C. After five washes with PBST (PBS containing 0.05% Tween-20), the membrane was probed with HRP-labeled Goat anti-Mouse IgG (H + L) Secondary Antibody (Thermo Fisher Scientific) at a dilution of 1:5000 for 1 h at room temperature. After being thoroughly washed with PBST, the target protein blot on the membrane was developed using an enhanced chemiluminescence detection kit (Thermo Fisher Scientific). The images were taken using a ChemiDoc^TM^ MP Imaging System (Bio-Rad Laboratories, Hercules, CA, USA).

### 2.5. IFA

LLC-PK1, ST, MA104 and Vero cells grown to ~80% confluence in 96-well cell plates were infected with PDCoV, TGEV, PRoV, and PEDV or PEAV, respectively, at a multiplicity of infection (MOI) of 1. After 1 h of adsorption at 37 °C, the inocula were discarded, and fresh DMEM containing 10 μg/mL of trypsin was added to the cells. At 48 h post-infection (hpi), the cells were fixed with 4% paraformaldehyde and then permeabilized with 0.3% Triton X-100. After being washed with PBS, the cells were blocked with 5% bovine serum albumin (BSA) for 30 min at room temperature and then incubated with either swine sera (1:100 dilution) or mAb 1A3 (1:40 dilution) for 1 h at 37 °C. Following three washes with PBS, the cells were incubated with FITC-conjugated Rabbit anti-Pig IgG (H + L) or Goat anti-Mouse IgG (H + L) for 1 h at 37 °C. After being washed thrice with PBS, the cells were counterstained with 4′,6-diamidino-2-phenylindole (DAPI) for 5 min at room temperature. After a final rinse, the cells were observed with an Eclipse Ci-S fluorescence microscope (Nikon Corp., Tokyo, Japan).

### 2.6. Establishment of PDCoV N-Based Blocking ELISA

A checkerboard titration was used to determine the optimal antigen-coating concentration and working concentration of the mAb 1A3 using the two reference swine sera S23 and S0. Specifically, 96-well ELISA plates (Corning Inc., Kennebunk, ME, USA) were coated with 100 μL/well of 2-fold serially diluted PDCoV N protein in the range of 0.15625–5 µg/mL for 12 h at 4 °C. After being washed three times with PBST, the plates were blocked with 2% BSA diluted in PBST for 2 h at 37 °C. After another three washes with PBST, 100 μL/well of undiluted positive and negative reference swine sera was separately added to the odd and even columns of the plates, which were then incubated for 1 h at 37 °C. After washing, 100 μL/well of the 2-fold serial dilutions of PDCoV N-specific mAb 1A3 (supernatant of the hybridoma cells) ranging from 1:10 to 1:320 was added to each well. The plates were then incubated for 1 h at 37 °C. Upon washing, 100 μL/well of HRP-conjugated Goat anti-Mouse IgG at 1:4000 dilution was added to each well, and the plates were incubated for 1 h at 37 °C. After a final wash, 100 μL/well of 3,3′,5,5′-tetramethylbenzidine (TMB) substrate was added to the plates for a 10 min chromogenic reaction at room temperature in the dark. After stopping the reaction by adding 50 μL/well of 2 M H_2_SO_4_, the optical density at 450 nm (OD_450_) of each well was determined using a Multiskan Sky Microplate Spectrophotometer (Thermo Fisher Scientific). The combination that produced the highest OD_450_ ratio of positive to negative reference serum (P/N) and an OD_450_ of the positive serum closest to 1.0 was defined as the optimal N protein-coating concentration and 1A3 working concentration. Furthermore, other key parameters that might affect the performance of the blocking ELISA, including distinct blocking buffers (2% BSA, 5% skimmed milk and 0.4% gelatin), HRP-conjugated Goat anti-Mouse IgG dilutions (1:2000, 1:4000 and 1:8000), and antigen-antibody reaction times (0.5, 1.0, 1.5 and 2 h), were also optimized under the optimal coating-antigen and mAb concentrations. Those conditions that produced the maximum P/N ratio were selected as the optimal parameters for the blocking ELISA. The reduction in OD_450_ value caused by serum antibodies blocking mAb binding was calculated for each sample using the following formula: percent inhibition (PI) = [(OD_450_ value of negative reference serum − OD_450_ value of tested serum)/(OD_450_ value of negative reference serum − OD_450_ value of positive reference serum)] × 100%.

### 2.7. Determination of Cut-Off Value, Diagnostic Specificity and Diagnostic Sensitivity

The cut-off value of the developed N-based blocking ELISA was determined using 384 swine serum samples collected from individual piglets with known PDCoV antibody status, which was confirmed using the reference method—IFA. The detection results of the blocking ELISA were compared with those obtained from the IFA. Receiver operating characteristic (ROC) analysis was conducted to analyze the blocking ELISA results obtained with the positive- and negative-testing serum samples in order to determine an optimized cut-off value that maximizes both the diagnostic specificity and diagnostic sensitivity of the assay. The PI value of each serum was analyzed by the ROC curve using GraphPad Prism software (Version 5.0; La Jolla, CA, USA) to present the area under the curve (AUC) at a 95% CI. Furthermore, the IFA test result of each serum sample and its corresponding PI value were also analyzed by the ROC curve using IBM SPSS 26.0 software (Armonk, NY, USA) to present the corresponding diagnostic sensitivity and specificity values constituting the ROC curve. The resulting sensitivity and specificity values were then used to calculate the Youden index, defined as max*_c_*{Sensitivity(*c*) + Specificity (*c*) − 1} (*c* stands for a threshold value), which is one of the most accurate methods for the calculation of the optimal cut-off point for diagnostic methods [29,30]. The PI cut-off value of the developed blocking ELISA was set at such a numerical parameter, below which all the serum samples tested negative via the IFA would fall into the negative range of the developed blocking ELISA, thereby being diagnosed as PDCoV seronegative by the blocking ELISA.

### 2.8. Analytic Sensitivity Evaluation of the Blocking ELISA

Two-fold serially diluted PDCoV antibody-positive reference sera in the range of 1:8–1:1024 were used to evaluate the analytical sensitivity of the developed blocking ELISA. One hundred microliters of each dilution were tested in octuplicate via the ELISA under the optimal conditions. The maximum dilution of the reference antiserum that still produced a PI value greater than the positive cut-off point was defined as the analytic sensitivity of the blocking ELISA. For comparison, IFA was performed in parallel on the same dilutions of the antiserum. Furthermore, an in-house indirect ELISA developed to detect PDCoV antibodies, using the recombinant PDCoV N protein as the coating antigen and paired with a cut-off value of 0.290 [28], was also performed in octuplicate on each of the antiserum dilutions. The resulting data sets were used to calculate the 95% limit of detection (LOD) via Probit regression analysis using SPSS 26.0 software (IBM Corp., Armonk, NY, USA) as we previously described [31].

### 2.9. Analytic Specificity Evaluation of the Blocking ELISA

Firstly, the specificity of the mAb 1A3 used in the blocking ELISA was analyzed via IFA to see whether it cross-reacted with other important pathogens causing diarrhea in pigs, including PEDV, TGEV, PRoV, and PEAV. Subsequently, the analytic specificity of the developed blocking ELISA was evaluated by testing its cross-reactivity with the antisera against PRoV, PEDV, TGEV, and PEAV. Three replicates of each antiserum were run on the same occasion under the optimal conditions. Moreover, for an analytical specificity comparison, the foregoing antisera were also tested using the in-house indirect ELISA [28].

### 2.10. Repeatability and Reproducibility Evaluation of the Blocking ELISA

To evaluate the intra-assay repeatability and inter-assay reproducibility of the developed blocking ELISA, three PDCoV antisera representing strongly, moderately and weakly positive samples were tested using the ELISA on one plate in one run or on three distinct plates in three independent runs. Each serum was tested in triplicate. The coefficient of variation (CV) was used to quantify the degree of variation of the blocking ELISA, which was calculated by dividing the standard deviation (SD) by the mean PI value of each antiserum.

### 2.11. Data Analysis

Software of GraphPad Prism and SPSS 26.0 were used to perform ROC analysis and Youden’s J statistic.

## 3. Results

### 3.1. Expression, Purification and Identification of A Recombinant PDCoV N Protein

The recombinant plasmid pET-28a-PDCoV-N was transformed into *E. coli* Rosetta (DE3) competent cells, which were then induced with 1 mM IPTG for 6 h at 37 °C. The expressed PDCoV N protein was analyzed via SDS-PAGE and Western blot analyses. As shown in Figure 1A, a thick protein band around 42 kDa, which is consistent with the theoretical molecular weight of the recombinant PDCoV N protein, was detected both in the supernatants and in the precipitates of bacterial lysates. Since more protein was expressed in the soluble form, we chose to purify the PDCoV N protein using Ni-NTA agarose (Qiagen) under native conditions. After purification, a clear protein band with a molecular weight of about 42 kDa was detected (Figure 1A), and its protein concentration was determined to be 0.5 mg/mL. Western blot analysis further demonstrated that the purified PDCoV N protein could be specifically recognized by an anti-His mAb (Figure 1B) and the mAb 1A3 raised against the PDCoV N protein (Figure 1C).

### 3.2. Establishment and Optimization of PDCoV N-Based Blocking ELISA

The blocking ELISA to be developed was designed using swine serum as the primary antibody and an anti-N mAb as the detection antibody. Based on the checkerboard titration results, the combination of 0.625 µg/mL of recombinant PDCoV N protein and 1:40 dilution of mAb 1A3 (supernatant of the hybridoma cells) produced the maximum P/N OD_450nm_ ratio and was thus selected as the optimal antigen-coating concentration and detection antibody dilution of the developed blocking ELISA (Figure 2A). Furthermore, other important conditions of the blocking ELISA, including blocking buffers, HRP-conjugated Goat anti-Mouse IgG and immunoreaction times, were also optimized. Figure 2B shows that 2% BSA in PBST exhibited the best coating effect among the three blocking buffers we tested; Figure 2C shows that the optimal working dilution of the HRP-conjugated secondary antibody was 1:4000; and Figure 2D shows that the optimal immunoreaction time for both serum samples and the HRP-conjugated secondary antibody was 1 h.

### 3.3. Determination of the Optimal Cut-Off Value, Diagnostic Specificity and Sensitivity

After optimizing the major reaction conditions of the blocking ELISA, a total of 384 swine serum samples were used to evaluate the performance of the assay. Before testing with the blocking ELISA, all swine serum samples were analyzed via IFA to confirm the PDCoV seronegative and seropositive status. Upon detection via IFA, 330 and 54 serum samples were confirmed to be positive and negative for PDCoV antibodies, respectively. Subsequently, all the samples were tested using the developed blocking ELISA, and the PI value of each sample was calculated. A ROC curve statistical analysis was performed to determine the PI cut-off value and to evaluate the diagnostic sensitivity and specificity of the assay. As shown in Figure 3A, the AUC was determined to be 0.9993 (*p* < 0.0001) with a 95% CI of 0.9982 to 1.000, indicating that the blocking ELISA had a significant coincidence rate with IFA. With the help of SPSS software, the diagnostic sensitivity and specificity values constituting the ROC curve were presented (data not shown), and it was shown that only when the diagnostic sensitivity and specificity were 98.79% and 100.00%, respectively (Figure 3A), was the corresponding Youden index (0.9879 + 1 − 1 = 0.9879) the maximum, which corresponded to a PI cut-off value of 51.65% for the developed blocking ELISA (Figure 3B). Based on the cut-off value of 51.65%, of the 330 serum samples tested positive via IFA, 326 and 4 were tested PDCoV seropositive and seronegative in the blocking ELISA, respectively, and of the 54 serum samples tested negative by IFA, all were tested PDCoV seronegative in the blocking ELISA (Table 1). Compared with the IFA detection results, the diagnostic sensitivity and diagnostic specificity of the developed blocking ELISA were 98.79% (326/330) and 100.00% (54/54), respectively. The overall coincidence rate between the two assays was 98.96% (380/384). In addition, it should be pointed out that the serum samples whose PI values were near the cut-off value (51.65%) of the blocking ELISA were mainly collected during 5–10 dpi (Figure 3B), which is consistent with previous studies demonstrating that PDCoV-specific antibodies are detectable in the serum at 7–14 dpi [8,32].

### 3.4. Analytic Specificity of the N-Based Blocking ELISA

Before assessing the analytic specificity of the developed blocking ELISA, we first analyzed the specificity of the mAb 1A3 which would be used as the detection antibody in the blocking ELISA. As shown in Figure 4, the mAb 1A3 only reacted with PDCoV-infected LLC-PK1 cells, manifesting as an obvious cytoplasmic immunofluorescent staining signal, which resembled that of the positive reference antiserum against PDCoV. No immunofluorescent staining signal was observed in TGEV-, PRoV-, PEDV- and PEAV-infected cells, indicating that the mAb has excellent specificity. On this basis, we further analyzed the analytic specificity of the blocking ELISA by determining its cross-reactivity with the antisera against other important viruses causing diarrhea in pigs, such as PEDV, TGEV, PRoV and PEAV. Figure 5 shows that only PDCoV antiserum was tested positive, while the antisera against PEDV, TGEV, PRoV, and PEAV were tested negative via the developed blocking ELISA. These results indicate that the developed blocking ELISA is specific for the detection of PDCoV antibodies. Furthermore, the in-house indirect ELISA yielded the same analytical specificity when detecting the foregoing antiserum samples (data not shown).

### 3.5. Analytic Sensitivity of the N-Based Blocking ELISA

Two-fold serial dilutions of the positive reference swine serum S23 in the range of 1:8–1:2048 were simultaneously detected using the developed blocking ELISA and the IFA. As shown in Figure 6, a dilution of 1:512 was the highest serum dilution producing positive test results in the blocking ELISA. In contrast, the highest serum dilution that was tested positive via the IFA was 1:256 (data not shown). Moreover, probit regression analyses further demonstrated that the 95% LOD of the blocking ELISA and indirect ELISA was 1:396 and 1:339, respectively. Hence, the analytical sensitivity of the blocking ELISA is slightly higher than that of the indirect ELISA, though only by a small degree.

### 3.6. Repeatability and Reproducibility of the N-Based Blocking ELISA

In the repeatability analysis, three strongly positive, medium positive and weakly positive PDCoV antisera were tested using the developed blocking ELISA on one plate in one run. In the reproducibility analysis, the three antisera were tested using the blocking ELISA on three different plates in three independent runs. As shown in Table 2, the intra-assay CV ranged from 2.38% to 4.18%, while the inter-assay CV ranged from 4.66% to 6.54%, indicating good repeatability and reproducibility of the blocking ELISA. These results indicate that the developed blocking ELISA is suitable for the serological diagnosis of PDCoV.

## 4. Discussion

As a newly discovered coronavirus [4], PDCoV has been proven not only to seriously threaten the world’s pig industry [33] but also to infect a variety of mammalian and avian species [15,34]. More worryingly, PDCoV has recently been demonstrated to be able to infect humans and is associated with an acute undifferentiated febrile illness in Haitian children [16]. This evidence indicates that PDCoV not only has the ability of interspecific transmission and zoonotic risk, but also has a wide host spectrum. At present, whether yet unidentified PDCoV hosts exist or whether the virus will spill over into new hosts remains unclear and warrants further scientific investigation [35,36]. Given that no effective therapeutic agent or commercial vaccine is currently available to treat or prevent PDCoV infection [37], rapid and reliable diagnosis will help to take disease management measures in time and initiate an epidemiological investigation of PDCoV infection.

As one of the most commonly used serological assays, the ELISA is capable of identifying a previous exposure to a pathogen, evaluating the kinetics of antibody response to the pathogen and facilitating the progression of an epidemiological investigation of the infection. Although a series of N-, S1-, and M-based indirect ELISAs have been established and tried to be applied to diagnose PDCoV infection in pigs [21,24,25,26], these assays are not applicable to the detection of samples of animal species other than pigs, because usually, only one kind of HRP-conjugated secondary antibody against one animal species is provided in the same ELISA kit. To overcome this problem, we made an attempt to develop a blocking ELISA by introducing an N-specific mAb 1A3 [8], which was used to compete with the virus-specific serum antibodies to bind to the coating antigen. Although this format of ELISA only contained one kind of HRP-conjugated secondary antibody against mouse, it is able to detect PDCoV antibodies derived from various animal species and human beings. However, due to the lack of PDCoV antisera against other animal species, we did not verify the actual effect of this blocking ELISA on the detection of samples from diverse animal species in this study. Instead, we used our developed blocking ELISA to test a polyclonal antibody against the PDCoV N protein produced in rabbits [27]. As expected, the polyclonal antibody was tested positive via the blocking ELISA (data not shown), indicating that the assay is able to detect PDCoV antibodies of various species origins. Another major advantage is that, with the introduction of mAb 1A3, the diagnostic specificity of the developed blocking ELISA will undoubtedly be improved, thus reducing false positives during clinical diagnosis. The antigenic epitope recognized by 1A3 was mapped to be located at amino acids 276–285 (RLKDALNTVV) of PDCoV N protein. Additionally, pLogo analysis further demonstrated that the 10 amino acids constituting the epitope were highly conserved among the 125 global PDCoV strains available in the GenBank database at the time of study initiation (unpublished data).

Furthermore, it is necessary to point out that although the N, S1 and M proteins of PDCoV can all be used as the diagnostic target, and indirect ELISAs based on these proteins have indeed been established [21,24,25,28], we found that the N protein exhibits the highest conservation level among the three proteins through amino acid sequence alignment analysis of 125 global PDCoV strains. The amino acid homologies of N, S1 and M proteins among the 125 PDCoV strains were 96.94–100.00%, 95.16–100.000% and 95.72–100.100%, respectively (Appendix A). Therefore, we chose to use the N protein as the diagnostic target for the blocking ELISA.

In order to obtain the optimal PI cut-off value for our developed blocking ELISA, a panel of 384 swine serum samples collected from piglets experimentally infected with PDCoV were simultaneously detected via the ELISA and IFA, which was used as the reference method to confirm the PDCoV antibody status of all serum samples. The resulting PI values of the 384 sera were then used to calculate the optimum cut-off point via ROC curve analysis in conjunction with Youden’s J statistic [29]. The Youden’s index is defined based on all points of a ROC curve, and the maximum value of the index corresponded to the optimum cut-off point of a diagnostic assay [29]. By calculation, the optimal PI cut-off value of the blocking ELISA was determined to be 51.65%. In the diagnosis of the 384 serum samples, four IFA-positive/ELISA-negative and no IFA-negative/ELISA-positive serum samples were identified, which corresponded to a diagnostic sensitivity of 98.79% (326/330) and a diagnostic specificity of 100.00% (54/54), respectively, for our developed blocking ELISA. These results indicate that the IFA exhibited a slightly higher sensitivity than our developed blocking ELISA in the detection of PDCoV antibodies. This is not an unanticipated result because PDCoV-infected cells were used as the antigen matrix to capture the serum antibodies, and any antibodies against different proteins of PDCoV produced in the serum would be recognized by PDCoV-infected cells in the IFA test. As a consequence, the IFA exhibited relatively higher sensitivity. Nonetheless, the IFA still has some disadvantages compared with the blocking ELISA we developed. On the one hand, both cell culture and virus inoculation are time-consuming and require skilled personnel. If the cells are in poor condition or the infectious dose of PDCoV is inappropriate, this will adversely affect the IFA test results of the serum samples. For example, if the viral infectious dose is too high, the cells will collapse and detach from the cell plate, resulting in false-positive IFA test results. On the other hand, the interpretation of the IFA test results is usually subjective and requires experienced personnel, especially when testing weakly positive serum samples. Moreover, it is difficult to commercialize the IFA, which constrains its wide use in PDCoV clinical diagnosis, especially in resource-limited diagnostic laboratories.

Compared with the IFA, although the blocking ELISA we developed exhibited a slightly lower diagnostic sensitivity, its main advantages include fast detection speed, easy performance, cost effectiveness, high-throughput detection, and easy commercialization. Undoubtedly, our developed blocking ELISA is a robust tool for the detection of PDCoV antibodies.

## 5. Conclusions

By preparing a recombinant PDCoV N protein and its specific monoclonal antibody, a blocking ELISA was developed to detect PDCoV antibodies. The diagnostic sensitivity and specificity of the ELISA were determined using 384 swine serum samples with known PDCoV antibody statuses, which were confirmed using the reference method—IFA. The blocking ELISA exhibited excellent repeatability in the detection of PDCoV antibodies and did not cross-react with antisera against other pathogens causing diarrhea in pigs. To our knowledge, this is the first report on the development of a blocking ELISA for PDCoV serodiagnosis.

## Figures and Tables

**Figure 1 viruses-14-01815-f001:**
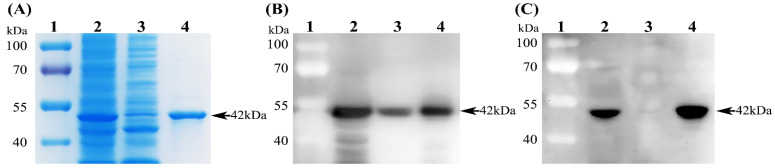
Identification of the recombinant PDCoV N protein. (**A**) SDS-PAGE analysis of the re-combinant PDCoV N protein. (**B**) Western blot identification of the recombinant PDCoV N protein using a mouse anti-His monoclonal antibody (mAb). (**C**) Western blot identification of the recom-binant PDCoV N protein using the mAb 1A3. Lane 1, Prestained Protein Ladder; Lane 2, super-natants of the bacterial lysates; Lane 3, precipitates of the bacterial lysates; Lane 4, the purified PDCoV N protein.

**Figure 2 viruses-14-01815-f002:**
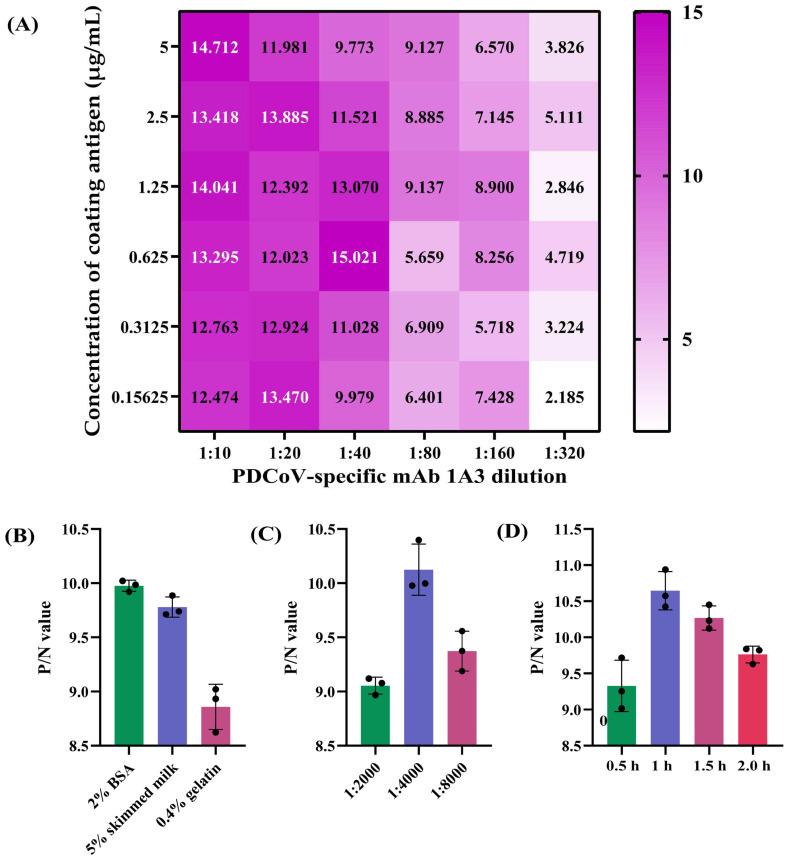
Optimization of the developed PDCoV N-based blocking ELISA. (**A**) Determination of the optimal working concentration of coating antigen and mAb 1A3 by checkerboard titrations. Theoptical density at 450 nm (OD_450nm_); ratios of positive to negative reference serum (P/N) are presented in a heatmap, which was drawn with the GraphPad Prism software. The darker the color, the greater the P/N OD_450nm_ ratio. (**B**) Comparison of the blocking effect of three blocking buffers (2% bovine serum albumin (BSA), 5% skimmed milk and 0.4% gelatin). (**C**) Determination of the optimal working dilution of the horseradish peroxidase (HRP)-conjugated Goat anti-Pig IgG secondary antibody. (**D**) Determination of the optimal immunoreaction time for both serum samples and HRP-conjugated secondary antibody. All results were presented as the mean ± SD of triplicate experiments. The black circles represent the P/N values of three independent runs.

**Figure 3 viruses-14-01815-f003:**
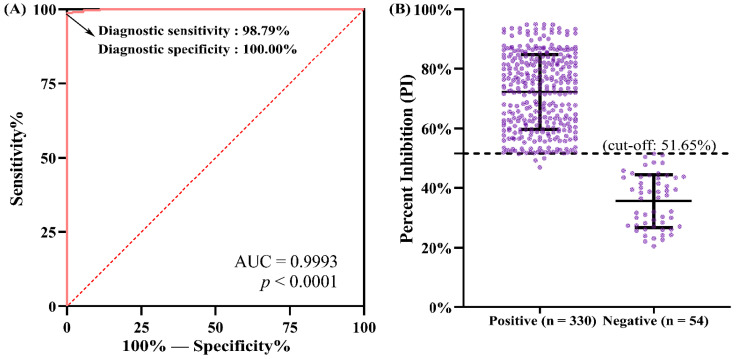
Cut-off value determination, diagnostic sensitivity and specificity analysis of the PDCoV N protein-based blocking ELISA. Receiver operating characteristic (ROC) analysis was performed using 384 swine serum samples with known PDCoV antibody statuses, including 330 positives and 54 negatives. (**A**) ROC curve showing the accuracy value was reflected by the area under the curve (AUC). ROC analysis was performed using GraphPad Prism software (Version 5.0; La Jolla, CA, USA). (**B**) Interactive plot showing the optimal percent inhibition (PI) cut-off value and diagnostic sensitivity and specificity. The optimal cut-off value for the blocking ELISA was calculated using the Youden index.

**Figure 4 viruses-14-01815-f004:**
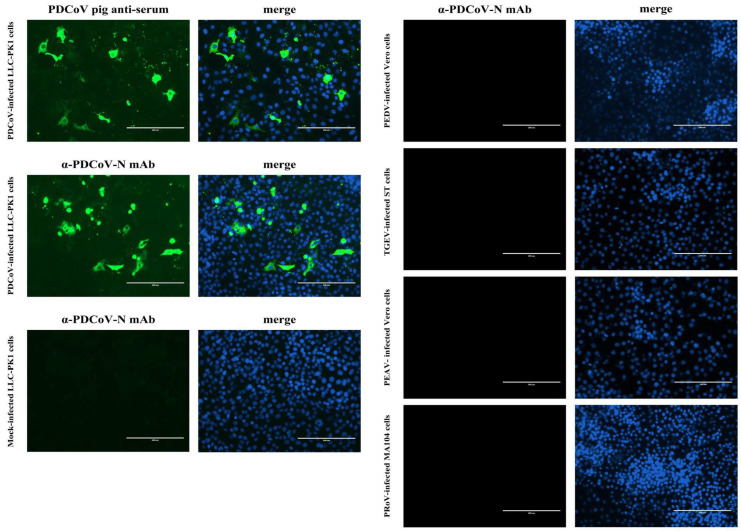
IFA analysis of the mAb 1A3 raised against the PDCoV N protein. LLC-PK1, ST, MA104, and Vero cells were mock-infected or infected with PDCoV, TGEV, PRoV, and PEDV or PEAV, respectively, at a multiplicity of infection of 1 for 48 h post-infection. The cells were fixed and processed for immunostaining with monoclonal antibody1A3 or PDCoV antiserum, followed by staining with fluorescein-isothiocyanate-conjugated Goat anti-Mouse IgG or Rabbit anti-Pig IgG. Cell nuclei were counterstained with 4′,6-diamidino-2-phenylindole. Scale bars, 200 µm.

**Figure 5 viruses-14-01815-f005:**
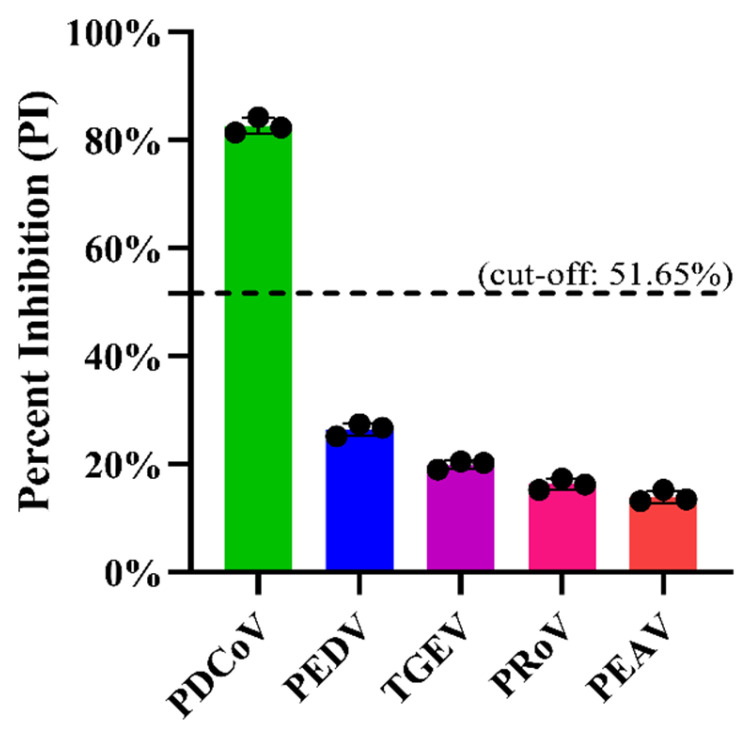
Analytic specificity of the developed PDCoV N-based blocking ELISA. Antisera against PDCoV, PEDV, TGEV, PRoV and PEAV were tested via the ELISA. The PI cut-off value of 51.65% was marked with a dashed line. The black circles represent the PI values of three independent runs on each antiserum.

**Figure 6 viruses-14-01815-f006:**
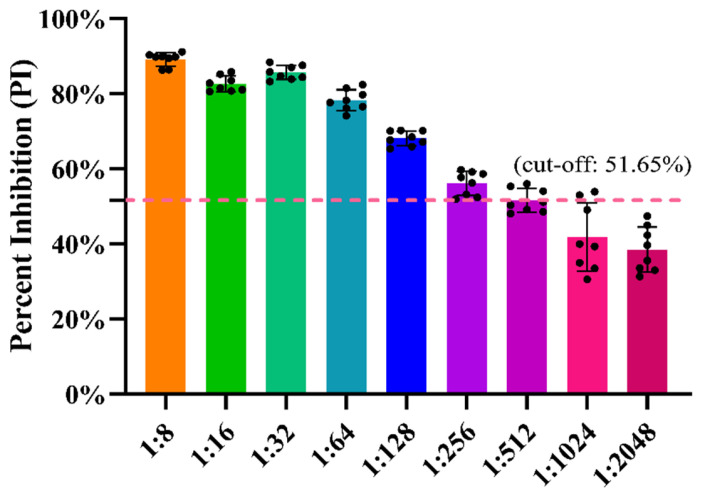
Analytic sensitivity of the developed PDCoV N-based blocking ELISA. Two-fold serially diluted positive reference PDCoV antiserum ranging from 1:8 to 1:2048 were detected via the ELISA. The PI cut-off value of 51.65% was marked with a dashed line. The black circles represent the PI values of eight independent runs on each dilution.

**Table 1 viruses-14-01815-t001:** Comparison of the developed blocking ELISA with IFA for the detection of PDCoV antibodies on 384 swine serum samples.

Assay		Blocking ELISA		Total	Kappa	*p*-Value
		Positive	Negative			
**IFA**	Positive	326	4	330	0.958	<0.001
	Negative	0	54	54		
	Total	326	58	384		

**Table 2 viruses-14-01815-t002:** Repeatability and reproducibility analysis of the developed PDCoV N-based blocking ELISA.

PDCoV Antisera	Repeatability (Intra-Assay)			Reproducibility (Inter-Assay)		
	Mean	SD	CV	Mean	SD	CV
Strongly positive	84.91%	0.035	4.18%	86.90%	0.057	6.54%
Moderately positive	76.84%	0.027	3.56%	79.44%	0.037	4.66%
Weakly positive	67.53%	0.016	2.38%	68.45%	0.035	5.11%

Mean: the average of PI values from three repeated blocking ELISA detections; CV: coefficient of variation; SD: standard deviation.

## Data Availability

Not applicable.

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
