# Peer review of "Development of a Nucleocapsid Protein-Based Blocking ELISA for the Detection of Porcine Deltacoronavirus Antibodies"

_viruses, 2022, doi:10.3390/v14081815_

Round 1

Reviewer 1 Report

This manuscript reports the newly developed ELISA for detecting porcine deltacoronavirus antibodies. The authors used 330 serum samples positive for anti-PDCoV antibodies and 54 serum samples negative for the antibodies to determine the conditions. The authors confirmed that overall coincidence rate between IFA and rate between IFA and established ELISA was 98.96%. Furthermore, there was no cross-reatioin with antisera against other swine pathogens. The study is well-designed and the manuscript is well-written. However, the advantages of “blocking ELISA” remains unclear in the manuscript. Thus, this reviewer recommends some following minor revisions.

1.                       Although the authors claimed that they established “Blocking ELISA”, it seems to be “competitive ELISA” which has already been developed. The authors should rephrase “Blocking ELISA”.

2.                       Furthermore, since HRP-conjugated anti-pig IgG antibody is available, standard ELISA methods (indirect ELISA method) is applicable. The authors should compare the sensitivity and specificity between the indirect ELISA and competitive ELISA methods.

3.                       Although the authors claimed that competitive ELISA is applicable for various species in Discussion (lines 407-412), there is no actual data to support this statement in the manuscript. Is epitope of mAb 1A3 conserved among various PDCoVs isolated from other species? Antigens and detecting antibodies would be need to be modified for serological study for other species.

4.                        “3.1. Construction of Recombinant Plasmid…” (lines 251-257) should move to Materials and Methods section.

5.                       Figure 2A: unit for concentration (e.g., μg/ml) is missing.

Reviewer 2 Report

Based on the study on N protein is well-described work. However, it should be clarified why focus is given to N protein and why other viral proteins would not be considered to improve the efficacy, which is limited to N protein described here.

It would benefit readers to include future perspectives according to the current study outcome.

Reviewer 3 Report

In the manuscript, the results of the development and validation of an ELISA for detection of antibodies against PDCoV strains have been introduced.
It has not been described clearly if the isolated strains is the same to obteined in other states or countries?? I mean...The sequence of the strains used for antigen coating is the same in all sequences of PDCoV??.

No statement can therefore be made if the strain is that different from those usually detected in commercial farms.

How were the 324 field sera confirmed to be true antibody positive? if you don´t include negative sera???

Were acutely infected pigs sampled? Which age group has been sampled? For determination of the sensitivity of an ELISA, defined positive samples should be used.

Round 2

Reviewer 3 Report

I have read the response of authors but I cannot access the latest version of the article. If the observations I requested (which they describe) are included in the article, I don´t have objection to publishing the article.